



# A global perspective on atmospheric blocking using GPS radio occultation – one decade of observations

Lukas Brunner[1, 2] and Andrea K. Steiner[1, 2, 3]

[1]Wegener Center for Climate and Global Change (WEGC), University of Graz, Graz, Austria
[2]FWF-DK Climate Change, University of Graz, Graz, Austria
[3]Institute for Geophysics, Astrophysics, and Meteorology/Institute of Physics, University of Graz, Graz, Austria

*Correspondence to:* Lukas Brunner (lukas.brunner@uni-graz.at)

**Abstract.** Atmospheric blocking represents a weather pattern where a stationary high-pressure system weakens or reverses the climatological westerly flow at mid-latitudes for up to several weeks. It is closely connected to strong anomalies in key atmospheric variables such as geopotential height, temperature, and humidity. Here we provide, for the first time, a comprehensive, global perspective on atmospheric blocking and related impacts by using an observation-based data set from Global Positioning System (GPS) radio occultation (RO) from 2006 to 2016. The main blocking regions in both hemispheres and seasonal variations are found to be well-represented in RO data. The effect of blocking on vertically resolved temperature and humidity anomalies in the troposphere and lower stratosphere is investigated for blocking regions in the northern and southern hemisphere, respectively. We find a statistically significant correlation of blocking with positive temperature anomalies, exceeding 3 K in the troposphere and a reversal above the tropopause with negative temperature anomalies below -3 K in the lower stratosphere. Specific humidity is positively correlated with temperature throughout the troposphere with larger anomalies revealed in the southern hemisphere. At the eastern and equator-ward side of the investigated blocking regions, a band of tropospheric cold anomalies reveals advection of cold air by anti-cyclonic motion around blocking highs, which is less distinct in the southern hemisphere due to stronger zonal flow. We find GPS RO a promising new data set for blocking research giving insight into the vertical atmospheric structure, especially in light of the expected increase in data coverage that future missions will provide.

## 1 Introduction

Global weather and climate are determined by different processes such as the jet stream, the storm tracks, and blocking. Blocking is a particularly important feature in many regions at mid-latitudes (e.g., Woollings, 2010). It describes a synoptic situation, where a strong and stationary high pressure system weakens or reverses the climatological eastward flow at mid-latitudes (Rex, 1950; Trenberth and Mo, 1985; Tibaldi and Molteni, 1990; Pelly and Hoskins, 2003; Barriopedro et al., 2006; Croci-Maspoli et al., 2007; Oliveira et al., 2014). Due to its persistence of up to several weeks, atmospheric blocking significantly influences key atmospheric variables such as geopotential height (GPH), temperature, and humidity throughout the troposphere and lower stratosphere. Further impacts of blocking are surface extremes which can lead to severe damages on economy and society (e.g., García-Herrera et al., 2010; Gilbert, 2010; Rodrigues and Woollings, 2017).



In the northern hemisphere (NH) the main blocking regions are located over the North Atlantic and Europe (Euro-Atlantic blocking region) as well as over the North Pacific (also referred to as the Alaskan blocking region) (Barriopedro et al., 2010; Whan et al., 2016). The impact of blocking on surface temperature extremes is well-established for both regions and different seasons (e.g., Favre and Gershunov, 2006; Buehler et al., 2011; Pfahl and Wernli, 2012; Bieli et al., 2015; Whan et al., 2016;

Brunner et al., 2017). The connection to humidity, precipitation, and droughts has also been intensively investigated, especially in recent years (e.g., Carrera et al., 2004; Galarneau Jr. et al., 2012; Pfahl et al., 2015; Wise, 2016; Sousa et al., 2017).

In the southern hemisphere (SH) blocking occurs in the entire South Pacific between about 160°E and 75°W. Highest frequencies are found in the south-eastern Pacific during winter (e.g., de Adana and Colucci, 2005; Berrisford et al., 2007; Parsons et al., 2016). However, in the SH blocking occurrence is considerably lower than in the NH. Also, the impacts of

blocking on populated areas are weaker compared to the NH (e.g., Lejenäs, 1984; de Adana and Colucci, 2005). Due to this imbalance comparably few studies investigate blocking in the SH, mostly focusing on impacts in Australia and New Zealand (Australian-New Zealand blocking region) and in South America (south-eastern Pacific blocking region) (e.g., Marques and Rao, 1999; Cowan et al., 2013; Pook et al., 2013; Parker et al., 2014). Several studies have also looked into the influence of other phenomena like the El Niño-Southern Oscillation (ENSO) or the Antarctic Oscillation (AAO)/Southern Annular Mode

(SAM) on SH blocking (Damião Mendes and Cavalcanti, 2014; Oliveira et al., 2014).

The systematic and global detection and analysis of atmospheric blocking and its impacts sets demanding requirements to the data sets in use. Apart from global coverage, observations with high spatial and temporal resolution are needed. Hence blocking research is mainly relying on model output and reanalysis data rather than using direct observations. However, most models show only limited skill in blocking representation, as has been noted by many studies in the past (D'Andrea et al.,

1998; Vial and Osborn, 2012; Barnes et al., 2012; Anstey et al., 2013; Christensen et al., 2013; Dunn-Sigouin and Son, 2013; Masato et al., 2013). Recently, Davini and D'Andrea (2016) showed that current climate models still under-represent blocking occurrence by up to 50 %, particularly in the Euro-Atlantic blocking region. Reanalyses combine an atmospheric model with a range of observations from different measurement systems to approximate the atmospheric state as accurately as possible. Due to this data assimilation the accuracy of reanalyses is less well understood compared to observations (Parker, 2016). Also,

there can be significant differences between different reanalyses and the causes are not yet fully understood (Fujiwara et al., 2017). Brunner et al. (2016) demonstrated the potential of Global Positioning System (GPS) radio occultation (RO) to detect and analyze blocking in this observational data set, using two exemplary blocking cases in 2010 and 2013. GPS RO provides highly accurate measurements of atmospheric variables and has therefore the potential to complement models and reanalyses as data set for blocking research.

In this study we provide, for the first time, a global perspective on atmospheric blocking based on the RO record from September 2006 to August 2016 exploiting its good vertical resolution for investigating the atmospheric vertical structure in temperature and humidity during blocking events. In Sect. 2 we introduce the RO record as well as the reanalysis data sets used for comparison. Section 3 describes the blocking detection algorithm, the gridding method for RO as well as the computation of anomalies, composites, and significance testing. We present the results of our study in Sect. 4 and conclude with a summary

in Sect. 5.



## 2 Data

### 2.1 Radio occultation data

Global positioning system (GPS) radio occulation (RO) is an active limb-sounding technique (Kursinski et al., 1997; Hajj et al., 2002). The measurements are characterized by global coverage, high vertical resolution, high accuracy, and no need for inter-satellite calibration (e.g., Foelsche et al., 2011; Ho et al., 2012; Steiner et al., 2013). The resolution reaches about 60 km horizontally and 100 m vertically in the lower troposphere and about 300 km horizontally and 1.5 km vertically in the lower stratosphere (Melbourne et al., 1994; Kursinski et al., 1997; Gorbunov et al., 2004). RO data have, so far, been used for a range of different applications in monitoring atmospheric variability and changes in Earth's climate (Anthes, 2011; Steiner et al., 2011; Gleisner et al., 2015; Randel and Wu, 2015). Significant improvement of weather forecasting (e.g., Healy and Thépaut, 2006; Cardinali, 2009) and atmospheric reanalyses (e.g., Poli et al., 2010; Simmons et al., 2014) has been made since RO observations can be assimilated without bias correction and act as anchor measurements. Including RO into reanalyses can reduce biases in the troposphere and stratosphere in both hemispheres (Poli et al., 2010). Several studies also used RO data to investigate dynamical features of the atmosphere such as waves (Randel and Wu, 2005; de la Torre and Alexander, 2005; Tsuda, 2014), the ENSO (Scherllin-Pirscher et al., 2012; Sun et al., 2014), tropopause characteristics (Schmidt et al., 2008; Rieckh et al., 2014; Peevey et al., 2014; Randel et al., 2003; Schmidt et al., 2005), and blocking (Brunner et al., 2016).

In this study we use RO data processed by the Wegener Center occultation processing system version 5.6 (OPSv5.6). Quality-controlled measurements for the 10-year period from September 2006 to August 2016 are selected. A detailed description of the OPS retrieval is given by Schwärz et al. (2016, Appendix A therein). Error estimates are provided by Scherllin-Pirscher et al. (2017). The accuracy of the data is best in the upper troposphere and lower stratosphere with 0.7 K in temperature and 10 m in geopotential height for individual profiles (Scherllin-Pirscher et al., 2011a, 2017) and even better when averaging over a range of profiles (Scherllin-Pirscher et al., 2011b).

We compute daily fields at a regular $2.5° \times 2.5°$ grid using a weighted average in space and time applied to the randomly distributed RO events, following:

$$x_{\mathrm{grid}}(\lambda, \phi, d) = \frac{\sum_i w_i x_i(\lambda', \phi', d')}{\sum_i w_i}, \qquad (1)$$

where $x_{\mathrm{grid}}(\lambda, \phi, d)$ represents a certain grid cell centered at longitude $\lambda$, latitude $\phi$, and day $d$. Each RO event $x_i(\lambda', \phi', d')$ within $\pm 7.5°$ in longitude, $\pm 2.5°$ in latitude, and $\pm 2$ days of the grid cell center is considered and weighted with a Gaussian weighting function $w_i$. The weighting function is given as:

$$w_i = \exp\left(-\left[\left(\frac{\Delta\lambda}{L}\right)^2 + \left(\frac{\Delta d}{D}\right)^2\right]\right), \qquad (2)$$

with $\Delta\lambda = \lambda - \lambda'$, $\Delta d = d - d'$, $L = 7.5°$, and $D = 1$ day. This effective resolution has been chosen to minimize the number of empty grid cells while maintaining most of the atmospheric variability. For more detailed information on the applied gridding method we refer to Brunner et al. (2016).



**Table 1.** Summary of reanalysis products, their resolution, assimilation of GPS RO data, and reference publications.

| Name | Provider | Downloaded resolution | RO assimilation | Reference |
|------|----------|----------------------|-----------------|-----------|
| ERA-Interim | ECMWF | 6h, $2.5° \times 2.5°$ | since January 1st, 2001 | Poli et al. (2010); Dee et al. (2011) |
| JRA-55 | JMA | 6h, $1.25° \times 1.25°$ | since January 1st, 2001 | Ebita et al. (2011); Kobayashi et al. (2015) |
| MERRA-2 | NASA | 6h, $0.625° \times 0.5°$ | since July 15th, 2004 | McCarty et al. (2016); Gelaro et al. (in press) |

## 2.2 Reanalysis data

Different reanalyses have extensively been used to investigate blocking and to evaluate the model performance in blocking representation (e.g., Sinclair, 1996; Trigo et al., 2004; Sillmann et al., 2011; IPCC, 2013; Davini and D'Andrea, 2016; Schiemann et al., 2017). Here, we selected three reanalyses for comparison to RO: the European Centre for Medium-Range Weather
Forecasts (ECMWF) Reanalysis Interim (ERA-Interim), the Japanese 55-year Reanalysis (JRA-55) by the Japan Meteorological Agency (JMA), and the recently published second Modern-Era Retrospective analysis for Research and Applications (MERRA-2) by the National Aeronautics and Space Administration (NASA). We use GPH at the $500\,\mathrm{hPa}$ pressure level from September 2006 to August 2016, from ERA-Interim, JRA-55, and MERRA-2 for blocking detection. All three reanalyses have a native 6-hourly time resolution, which is averaged to daily fields. The varying spatial resolutions are interpolated to a
consistent $2.5° \times 2.5°$ longitude-latitude grid. Note that all three reanalyses also assimilate RO measurements as specified in Table 1.

## 3 Methods

A blocking detection algorithm based on the reversal of $500\,\mathrm{hPa}$ GPH gradients is applied to the RO data between September 2006 and August 2016. Resulting blocking frequencies are investigated with regard to their horizontal and temporal evolution
and compared to established reanalyses. Three main blocking regions in both hemispheres are selected and the vertical atmospheric structure of temperature and specific humidity anomalies during blocking in these regions is analysed. Statistically significant links between blocking and the anomalies in temperature and specific humidity are found via a Monte Carlo test.

### 3.1 Blocking detection in RO GPH fields

We use a standard $500\,\mathrm{hPa}$ GPH gradient algorithm (Tibaldi and Molteni, 1990; Scherrer et al., 2006; Davini et al., 2012,
2014), adapted to allow the simultaneous detection of blocking in the northern and southern hemisphere. First, GPH gradients





to the north ($\Delta Z_{\mathrm{N}}$) and to the south ($\Delta Z_{\mathrm{S}}$) are calculated for each grid cell :

$$\Delta Z_{\mathrm{N}}(\lambda, \phi) = \frac{Z(\lambda, \phi + \Delta\phi) - Z(\lambda, \phi)}{\Delta\phi} \tag{3}$$

$$\Delta Z_{\mathrm{S}}(\lambda, \phi) = \frac{Z(\lambda, \phi - \Delta\phi) - Z(\lambda, \phi)}{\Delta\phi}, \tag{4}$$

with the longitude $\lambda$ running from $180°$W to $177.5°$E and the latitude $\phi$ running from $72.5°$S to $72.5°$N. The gradient is calculated over a latitude width of $\Delta\phi = 15°$. By this definition the northern gradient $\Delta Z_{\mathrm{N}}$ is positive if the GPH is higher to the north and equivalently $\Delta Z_{\mathrm{S}}$ is positive if the GPH is higher to the south.

GPH-based blocking detection indices are usually restricted in latitude, to avoid the detection of low-latitude atmospheric waves which are not considered as blocking in the classical sense (e.g., Scherrer et al., 2006; Barriopedro et al., 2006; Martineau et al., in press). Particularly in hemispheric summer the pole-ward shift of slow-moving atmospheric ridges can otherwise lead to very high blocking frequencies equator-ward of $45°$ latitude (e.g., Davini et al., 2014). In order to avoid the detection of low-latitude blocking and for ensuring comparability of our results with existing literature, we introduce a third gradient towards the equator ($\Delta Z_{\mathrm{E}}$), following Davini et al. (2012):

$$\Delta Z_{\mathrm{E}}(\lambda, \phi) = \frac{Z(\lambda, \phi \mp 2 * \Delta\phi) - Z(\lambda, \phi \mp \Delta\phi)}{\Delta\phi} \quad \text{with} \begin{cases} - \text{ in the NH} \\ + \text{ in the SH,} \end{cases} \tag{5}$$

where the minus sign is valid in the NH and the plus sign is valid in the SH. Simply speaking, $\Delta Z_{\mathrm{E}}$ is defined positive at a certain grid cell if there is a clear trough in the GPH-field towards the equator and is used to prohibit the identification of slow moving low-latitude ridges as blocking.

Instantaneous blocking (IB) is identified on a grid cell basis if the following three conditions are simultaneously met:

$$\Delta Z_{\mathrm{N}}(\lambda, \phi) \begin{cases} < -10 \, \mathrm{m/deg.\,lat.} \text{ in the NH} \\ < 0 \, \mathrm{m/deg.\,lat.} \text{ in the SH} \end{cases} \tag{6}$$

$$\Delta Z_{\mathrm{S}}(\lambda, \phi) \begin{cases} < 0 \, \mathrm{m/deg.\,lat.} \text{ in the NH} \\ < -10 \, \mathrm{m/deg.\,lat.} \text{ in the SH} \end{cases} \tag{7}$$

$$\Delta Z_{\mathrm{E}}(\lambda, \phi) > 5 \, \mathrm{m/deg.\,lat.} \text{ for both hemispheres.} \tag{8}$$

We only consider IB events with an extent of at least $15°$ in longitude and filter out too small blocking systems. In a final step we define blocking for a given day and grid cell if such a large-scale event is also persistent and stationary, requesting IB to be found within a $10° \times 5°$ longitude-latitude region in the neighbouring $\pm 2$ days.

In addition, we define blocked days with respect to three selected regions. A blocked day is found if at least one grid point is blocked in such a region. The regions are chosen to cover the blocking maxima in both hemispheres and will, in the following, be referred to as the North Atlantic region ($30°$W to $10°$E and $30°$N to $72.5°$N), the North Pacific region ($160°$E to $160°$W and $30°$N to $72.5°$N), and the East Pacific region ($150°$W to $90°$W and $72.5°$S to $30°$S).



## 3.2 Anomaly computation in RO temperature and humidity fields

Anomalies of atmospheric temperature ($T_{\text{Anom}}$) and relative specific humidity ($q_{\text{Anom}}$) during blocked days $t$ are calculated for each location $(\lambda, \phi)$ and pressure level $p$:

$$T_{\text{Anom}} = T - \overline{T} \tag{9}$$

$$q_{\text{Anom}} = \frac{q - \overline{q}}{\overline{q}} \times 100\,\%, \tag{10}$$

with temperature $T = T(t, \lambda, \phi, p)$ and specific humidity $q = q(t, \lambda, \phi, p)$. Respective daily mean values $\overline{T} = \overline{T}(d, \lambda, \phi, p)$ and $\overline{q} = \overline{q}(d, \lambda, \phi, p)$ are calculated over the 10 years from September 2006 to August 2015 for each day of the year $d$. For specific humidity we show relative anomalies to allow easier comparison across different pressure levels due to its exponential decline with altitude. Composites of the temperature and specific humidity anomalies are then obtained by averaging over all blocked days $t$ of a certain region.

## 3.3 Statistical significance testing

Statistical significance of the composites is determined for each pressure level on a grid cell basis using a Monte Carlo test. Given $n$ blocked days in a certain region and period, 1000 samples of $n$ random days are drawn from the same period (e.g., season) and averaged. To conserve the autocorrelation, consecutive blocked days are clustered and lead to consecutive days in the random samples. Based on the 1000 random samples the probability density function (PDF) is calculated, with values below the 5th or above the 95th percentile of this PDF being considered statistically significant, respectively.

## 4 Results

### 4.1 Blocking climatologies from RO

Figure 1 shows annual blocking frequencies derived from the RO data set and the three reanalyses, ERA-Interim, JRA-55, MERRA-2. All four data sets agree on the two main blocking regions in the NH (Fig. 1, left panel). There is a clear maximum in the blocking frequency in the Euro-Atlantic blocking region between 50°W and 50°E and a smaller maximum in the North Pacific blocking region between 150°E and 150°W (compare IPCC, 2013, Box 14.2). In the Euro-Atlantic region the maximum frequency is between about 10 % for ERA-Interim and JRA-55 and about 10.5 % for MERRA-2, while the maximum RO frequency is a bit lower with 8 %. In addition, the RO maximum in this region is shifted by about 10° to the east compared to the reanalyses. All four data sets consistently place the minimum blocking frequency east of the Euro-Atlantic region at 100°E. RO shows frequencies of 2 % here, ERA-Interim and JRA-55 are about 1 % higher, and MERRA-2 is about 1.5 % higher. In the North Pacific region RO reaches a maximum frequency of about 6 %, while the reanalyses show about 7 % to 8 %. The region with lowest blocking frequencies below 1 % is found at 90°W across all data sets. In general, RO data show an underestimation of one-dimensional blocking frequencies. The absolute difference to the reanalyses stays below 2 % at most longitudes. Only near the maximum in the Euro-Atlantic blocking region the difference exceeds 3 %.





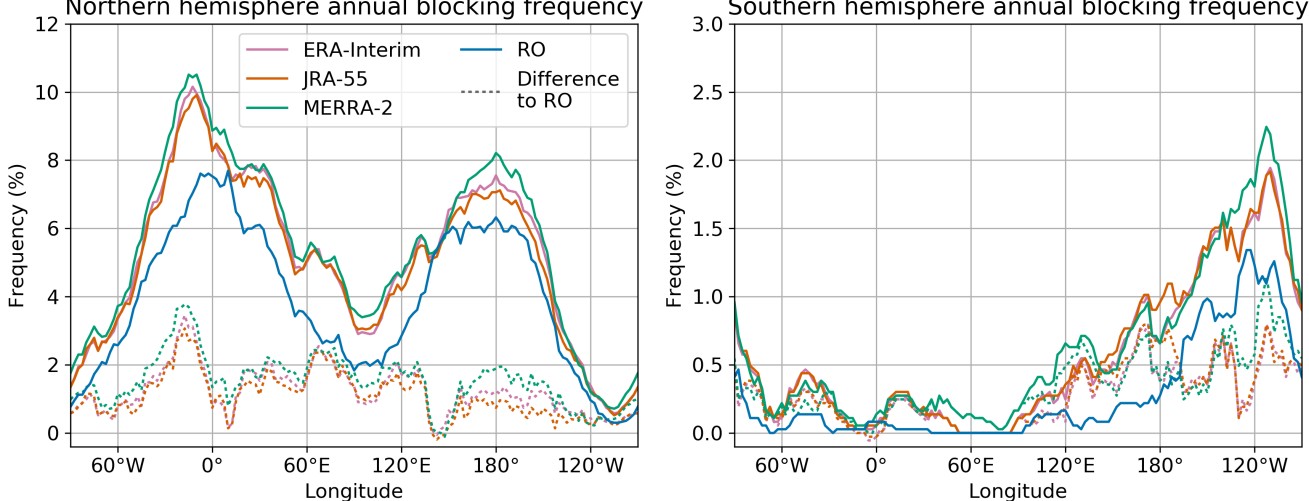

**Figure 1.** Annual blocking frequencies for the (left) northern and (right) southern hemisphere in the period 09/2006–08/2016. Each colored line represents a data set, the dashed lines show the respective differences of reanalyses to RO. Note the different y-axis ranges.

The right panel of Fig. 1 shows the SH blocking distribution. Again, all data sets agree on the main blocking region in the South Pacific. Highest frequencies are consistently found in the south-eastern Pacific between $150°$W and $100°$W. MERRA-2 shows the highest maximum frequency with about $2.25\%$, followed by ERA-Interim and JRA-55 with about $2\%$, and RO with about $1.75\%$. Eastward of the south-eastern Pacific region RO shows hardly any blocking and also all three reanalyses stay

below $0.5\%$ blocking frequency as well (corresponding to about two blocked days per year on average). In the Australian-New Zealand region between $100°$E and $150°$W RO blocking frequencies hardly exceed $0.5\%$ and the reanalyses hardly exceed $1\%$.

The time evolution of blocking is presented in Fig. 2 for both hemispheres from September 2006 to August 2016. Both main blocking areas in the NH, as well as the South Pacific region in the SH are clearly recognizable in this view. A closer

inspection reveals that a NH blocking has an average duration of 4 days and an average longitudinal extent of about $34°$. The most persistent blocking cases in the NH occurred in March/April 2007 (27 continuous days), in December/January 2009/2010 (28 days), and in February 2015 (23 days). In the SH an average blocking only lasts 2.5 days and has an extent of $23°$ in longitude. There the most persistent blocking cases are found in May/June 2012 (12 days), in July/August 2014 (8 days), and in September 2015 (8 days). In general, blocking in the SH is by far weaker and less frequent than in the NH.

Taking a closer look into the characteristic blocking features, we further investigate the distribution of blocking frequencies in longitude and latitude for different seasons. Figure 3 shows horizontally resolved blocking frequencies for all seasons in the NH for RO and ERA-Interim. A comparison of blocking frequencies with JRA-55 and MERRA-2 is not shown as they are highly consistent and agree within $0.2\%$ annual blocking frequency to ERA-Interim. RO resolves all the main features in the NH blocking distribution. Annual frequencies from RO show the main blocking regions over the North Atlantic and





**Figure 2.** Hovmöller diagram of blocking as function of time over longitudes for the (left) northern and (right) southern hemisphere based on the RO record for the period 09/2006–08/2016. Red arrows mark the three longest blocking events in each hemisphere.



**Figure 3.** Blocking frequencies for the northern hemisphere in the period 09/2006–08/2016. Frequencies are shown for RO (left), ERA-Interim (middle), and RO minus ERA-Interim (right) for (from top to bottom) the annual mean and seasonal means, spring, summer, fall, and winter.





Europe (Euro-Atlantic blocking region) as well as a maximum over the North Pacific. In the seasonally resolved analysis RO detects the highest blocking frequencies over the Euro-Atlantic region during winter (DJF) and spring (MAM) consistent with ERA-Interim. Blocking occurrence in the North Pacific region is high during the entire year, with fewest blockings in fall (SON). In general, RO and ERA-Interim agree very well on the location of the blocking regions in all seasons. Larger differences exceeding 2 % are only found in NH summer (JJA), where RO does not fully capture the frequency maxima over northern Russia. In winter RO shows slightly higher blocking frequencies than ERA-Interim in the North Atlantic and over Scandinavia.

One possible reason for the generally lower blocking frequencies in the RO record is the measurement density of the RO events. As described in Sect. 2, RO data are averaged from the randomly distributed measurements to a regular daily grid using a weighted mean. Due to the random distribution of observations it can happen that some grid cells do not have any contributing RO events (see also Brunner et al. (2016) for a detailed analysis). Such empty grid cells can artificially lower the blocking frequency if they appear at the location of a blocking.

We tested the effect of the weighted averaging in the gridding of RO data and applied the same weighted averaging in space and time to ERA-Interim data. Comparing then blocking frequencies from the simililarly weighted ERA-Interim fields to RO yields slightly reduced differences in blocking frequency and shows that about 0.5 % in difference can be explained by the weighted averaging.

In the SH the overall blocking frequency is notably lower compared to the NH. It has been argued that the stronger zonal flow at mid-latitudes in the SH leads to less persistent blocking conditions (e.g. Trenberth and Mo, 1985). Oliveira et al. (2014) suggest a three day stationarity criterion for blocking detection in the SH as opposed to the typical five day criterion in the NH to account for the stronger westerlies. However, we here aim at a consistent comparison of blocking in both hemispheres and therefore use the five-day criterion globally. This approach allows a direct comparison of blocking anti-cyclones and their impacts in both hemispheres.

In the SH blocking is almost exclusively found in the South Pacific (Fig. 4). Normally two sub-regions are distinguished mainly with regard to the impact on populated areas: blocking in the south-western Pacific (often referred to as the Australian-New Zealand blocking region) and blocking in the south-eastern Pacific region (influencing populated areas in South America; referenced to as the East Pacific region). In contrast to the NH, SH blocking is mainly constraint to the southern winter (JJA) season, where two-dimensional frequencies can reach 2 %. RO and ERA-Interim consistently show this seasonal development. Differences between the two data sets stay mostly below 0.25 % annually and below 0.5 % seasonally. Largest differences are found during the blocking maximum in SH winter.

## 4.2 Atmospheric temperature and specific humidity response to blocking

In the following we investigate the atmospheric structure of vertically resolved temperature and relative specific humidity anomalies in the troposphere and lower stratosphere during blocked days. The effects of blocking are shown for three regions (two in the NH, one in the SH) and for five selected pressure levels: 850 hPa, 500 hPa, 270 hPa, 200 hPa, and 100 hPa. These



**Figure 4.** As Fig. 3, but for the southern hemisphere. Note the different colorbar-ranges compared to Fig. 3.



**Figure 5.** Composites of (left) temperature anomalies and (right) relative specific humidity anomalies during blocked days in the North Atlantic region between 30°W–10°E and 30°N–72.5°N (grey box; 267 days in total). Shown is the northern hemisphere extended winter (NDJFM) season. Hatched regions denote statistical significance at the 5th/95th percentile level.





levels represent (bottom to top) regions of main blocking influence in the lower and middle troposphere region, the tropopause region, the region of main blocking influence in the lowest stratosphere and of decreasing influence in the stratosphere above.

Winter and summer seasons are compared in Figure 5 and 6 for temperature and relative specific humidity anomalies during blocked days over the North Atlantic region showing extended winter (NDJFM) and extended summer (MJJAS), respectively. During winter a clear and statistically significant positive temperature anomaly dominates most of the blocking region throughout the troposphere up to about 300 hPa. The anomalies reach about 2 K in the lower troposphere and exceed 3 K at their maximum at about 500 hPa. At lower pressures the positive anomalies decrease towards the tropopause. Beginning at about 300 hPa the decrease is accompanied by a shift to the north. The temperature anomalies are smallest at about 270 hPa, where they change from positive to negative. Higher up, in the lower stratosphere increasingly negative temperature anomalies, falling below -3 K at 200 hPa, are the dominating feature. Further up the influence of blocking on the temperature weakens and the anomalies decrease. A noticeable feature is also that the temperature anomalies are not centred in the blocking region in the troposphere near 500 hPa but appear to be shifted to the west. This asymmetry disappears at higher altitudes and especially the lower-stratospheric cold anomalies are perfectly centred in the blocking region.

In the troposphere the central positive temperature anomaly is surrounded by a cold anomaly on the northern, eastern, and southern flanks. This anomaly, which is considerably weaker in summer (cf., Fig. 6), hints at the influence of the circulation during blocked conditions. The anti-cyclonic motion of air around stationary high-pressure systems in the investigated region favours the advection of cold air from the north towards central Europe. The cold anomalies are stronger in the lower regions of the troposphere, falling below -2 K at 850 hPa. At 500 hPa a band of cold air with composite temperatures below -1 K is still visible to the east and south of the positive anomaly which change above the tropopause at the 200 hPa and 100 hPa level into positive anomalies of about 0.5 K to 1.5 K, especially north and south of the central cold anomaly.

The analysis of relative specific humidity anomaly composites (Fig. 5, right) reveals a clear correlation with temperature in most of the troposphere: positive temperature anomalies are accompanied by positive specific humidity anomalies and negative temperature anomalies are accompanied by negative specific humidity anomalies. However, dry anomalies are mostly restricted to the European continent, especially in the lower troposphere. In contrast to temperature, specific humidity anomalies do not change sign in the tropopause region. The strongest anomalies, exceeding 30 %, are found at the altitude of weakest temperature anomalies (at about 270 hPa). In the, generally, very dry stratosphere the specific humidity anomalies decrease rapidly and no statistically significant signal of blocking is found above about 150 hPa.

For extended summer, temperature and relative specific humidity anomaly composites during blocked days in the Euro-Atlantic region (Fig. 6) are about 1 K and 10 % lower compared to respective anomalies in winter. Moreover, the band of cold air surrounding the central warm anomaly is less distinct in summer. At 500 hPa, where the feature is clearest in winter, large regions, especially over north-eastern Europe, are not statistically significantly colder during blocked conditions. This indicates that cold advection from the north is less important during summer blocking.

Specific humidity anomalies during summer blocking in this region are not statistically significant in most of the troposphere. Stronger anomalies only appear near the tropopause and above. While in winter the strongest moist-anomalies are found at about 270 hPa, they appear higher up at about 200 hPa in summer.





**Figure 6.** As Fig. 5, but for the extended summer (MJJAS) season (186 days in total).






**Figure 7.** As Fig. 5, but for blocking in the North Pacific region between 160°E–160°W and 30°N–72.5°N (191 days in total).





In the North Pacific blocking region during extended winter (Fig. 7), the main feature in temperature is again a strong positive anomaly in the troposphere. Compared to the North Atlantic region the anomaly is stronger in the lower troposphere below 500 hPa, while the negative anomaly in the lower stratosphere is slightly weaker. The tropospheric cold anomalies are limited to east and southwest of the blocking region with the coldest temperatures found over the northwest of the North American continent. In the lower stratosphere the warm anomaly is limited to the south of the blocking region, creating a distinct dipole feature near the 200 hPa pressure level.

Specific humidity anomalies are strongest in the middle troposphere at about 500 hPa and about 10 % lower than in the North Atlantic region at lower pressures. At 270 hPa and 200 hPa a clear dipole similar to the 200 hPa temperature anomaly can be found. Above 200 hPa the influence of blocking on atmospheric humidity decreases and hardly any significant signal is found above.

For the SH (Fig. 8) we show blocking effects in the East Pacific region. Similar to the NH, both, temperature and relative specific humidity anomalies are clearly shifted to the west of the blocking region in the lower troposphere. The strongest temperature anomalies during blocking, clearly exceeding 3 K, are found in the lowermost part of the troposphere. Towards the tropopause the anomalies decrease and again change sign at about 270 hPa. The lowest temperature anomalies below -3 K are located at about 200 hPa, similar to the NH. Above, the influence of blocking on temperature decreases. The tropospheric cold anomalies, surrounding the blocking region are less distinct in the SH. These results suggest that cold advection plays a less important role in the SH due to the stronger zonal flow. A clear band of negative temperature anomalies is only visible at 500 hPa, while at 850 hPa the strongest cold anomalies are restricted to downstream of the blocking region. Compared to the NH a stronger second temperature maximum appears north-east of the blocking region.

Specific humidity anomalies in the SH show notably more variation than in the NH. Throughout the entire troposphere wet and dry anomalies exceed 30 %. The anomalies spread in a wave-like pattern from the blocking region to the northeast, which is most distinct near the tropopause at about 270 hPa. In the lower stratosphere the specific humidity anomalies again decrease rapidly.

In summary, we find similar effects of blocking on atmospheric temperature and specific humidity anomalies in the different investigated regions in both hemispheres. Largest differences in amplitude appear between the seasons, while the SH shows a more complex signature of blocking, especially in specific humidity. For all cases strong positive temperature anomalies are found in the lower to middle troposphere and a maximum negative anomaly in the lower stratosphere at about 200 hPa. Specific humidity anomalies are strongest higher up between 270 hPa and 200 hPa, except in the North Pacific region where the largest anomalies are found at the 500 hPa level.

# 5 Summary, Conclusions, and Outlook

We presented the first comprehensive analysis of global atmospheric blocking based on Global Positioning System (GPS) radio occulation (RO) observations. We used one decade of RO measurements from September 2006 to August 2016 to derive blocking climatologies and to investigate blocking impacts on vertically resolved atmospheric temperature and specific humid-





**Figure 8.** As Fig. 5, but for the southern hemisphere extended winter (MJJAS) shown for blocking in the East Pacific region between 150°W–90°W and 72.5°S–30°S (81 days in total).





ity fields. We investigated the representation of main blocking regions in the northern and southern hemisphere for different seasons. The impact of blocking on vertically resolved temperature and humidity was examined based on anomaly composites and its significance was tested.

Our results show that RO data are well suited for blocking detection. RO correctly resolves the blocking regions in both
hemispheres, also capturing the seasonal blocking variability. Average blocking episodes in the northern hemisphere (NH) are found to persist for 4 days and have a longitudinal extent of $34°$. In the southern hemisphere (SH) blocking is less persistent and lasts on average 2.5 days, with a typical extent of $23°$ in longitude.

The impact of blocking on temperature and specific humidity is found to be statistically significant throughout the troposphere and lower stratosphere in both hemispheres. During extended winter a strong positive temperature anomaly exceeding
$3\,\mathrm{K}$ is found in the center of the blocking area, slightly shifted to the east at lower altitudes. Above about $500\,\mathrm{hPa}$ this anomaly decreases until it changes sign above the climatological tropopause at about $270\,\mathrm{hPa}$. In the lower stratosphere, blocking leads to a negative temperature anomaly below -3 K at about $200\,\mathrm{hPa}$. Higher up the influence of blocking on temperature decreases. In the troposphere cold anomalies surround the central warm anomaly, indicating the effect of advection of cold air from the polar regions by the anti-cyclonic motion around blocking highs. In the lower stratosphere this anomaly also changes sign and
appears as anomalously warm region equator-ward of the block. Summer temperature anomalies are similar to those in winter, but notably weaker in amplitude of up to $50\,\%$. In addition, the advection of cold air plays a less important role, leading to less distinct negative anomalies in the troposphere.

Specific humidity anomalies show a similar behaviour as temperature in the troposphere. In the North Atlantic region, a central wet anomaly is surrounded by dry anomalies on the eastern and equator-ward side. However, the anomalies do not
change sign at the tropopause, leading to inverse patterns of temperature and specific humidity in the lower stratosphere. Above about $200\,\mathrm{hPa}$, the influence of blocking on specific humidity is found to decrease rapidly. In the southeast Pacific region, specific humidity anomalies are generally stronger than in the NH and show a wave-like pattern with positive and negative anomalies alternating from the southwest to the northeast, due to a stronger zonal flow.

Our findings highlight the main blocking regions in both hemispheres and the effect of blocking in these regions on atmo-
spheric temperature and specific humidity using GPS RO observations. The slight underestimation of blocking frequencies in RO compared to three different reanalyses, ERA-Interim, JRA-55, and MERRA-2, is most probably due to a too sparse measurement density. Future RO missions, like the FORMOSAT-7/COSMIC-2 constellation, and the exploitation of signals from more Global Navigation Satellite System (GNSS) constellations, like the European Galileo, the Russian GLONASS, and the Chinese BeiDou, are expected to significantly increase the number of RO events, promising to overcome this undersampling
and allowing an even better performance of RO data in blocking representation (Yue et al., 2014).

RO measurements provide a mostly independent, comprehensive observation-based record of known accuracy (Parker, 2016) for the detection and analysis of atmospheric blocking complementing reanalyses and models. The high vertical resolution of the RO measurements makes them ideal for investigating the atmospheric structure during blocking episodes. This will allow to gain a better understanding of the development of blocking related extreme events, like heat waves and cold spells, flooding,
or droughts, in the future.





## 6   Code availability

The analysis was carried out in Python 2.7, the code is available upon request from L. Brunner (lukas.brunner@uni-graz).

## 7   Data availability

We used geopotential height fields from three reanalyses: the European Centre for Medium-Range Weather Forecasts (ECMWF),

5    ERA-Interim: ECMWF Reanalysis Interim, accessed March 2017, available at http://apps.ecmwf.int/datasets/data/interim-full-daily/
levtype=pl/, the Japan Meteorological Agency (JMA), JRA-55: Japanese 55-year Reanalysis, accessed January 2017, available
at http://jra.kishou.go.jp/JRA-55/index_en.html, and the Global Modeling and Assimilation Office (GMAO) (2015), MERRA-
inst6_3d_ana_Np: 3d, 6-Hourly, Instantaneous, Pressure-Level, Analysis, Analyzed Meteorological Fields V5.12.4, Green-
belt, MD, USA, Goddard Earth Sciences Data and Information Services Center (GES DISC), accessed November 2016,

doi:10.5067/A7S6XP56VZWS, available at https://disc.gsfc.nasa.gov/.

The gridded RO data used in this study are available on request from L. Brunner (lukas.brunner@uni-graz) and will be
provided publicly soon.

*Author contributions.*   L. Brunner collected the data, performed the analyses, created the figures, and wrote the manuscript. Andrea K. Steiner
provided guidance on all aspects of the study and contributed to the text.

*Competing interests.*   The authors declare that they have no conflict of interest.

*Acknowledgements.*   The authors thank P. Davini for helpful advice and discussions and E. F. Hohmann for reminding us what is really
important. ECMWF, JMA, and NASA are acknowledged for providing their reanalyses as detailed in Sect. 7. UCAR/CDAAC (Boulder,
CO, USA) is acknowledged for access to its RO phase and orbit data, the WEGC processing team members (M. Schwärz, F. Ladstädter,
B. Angerer), are thanked for providing the OPSv5.6 RO data. This work was funded by the Austrian Science Fund (FWF) under research

grant W 1256-G15 (Doctoral Programme Climate Change – Uncertainties, Thresholds and Coping Strategies). L. Brunner was financially
supported by a Marietta Blau Grad by the Austrian Exchange Service (OeAD), financed by funds from the Austrian Federal Ministry of
Science, Research and Economy (BMWFW). The Center for International Climate Research Oslo (CICERO, Norway) is acknowledged for
hosting L. Brunner during part of the work. The Authors thank all contributors to Python and LaTeX, which were used throughout the work.



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
