# Peer review of "A global perspective on atmospheric blocking using GPS radio occultation – one decade of observations"

_Atmospheric Measurement Techniques, 2017_

## Referee Comment (RC1) · Anonymous Referee #1 · 10 Aug 2017

General comments: The paper is well-structured, making it easy for the reader to follow. Title and abstract are excellent as they reflect the content of the paper quite well. Figures have a high quality. The results are represented well, however, a detailed comparison to results of previous studies is missing, especially in the summary and conclusion part. In addition, there are some statements that should be clarified. The paper is written in a clear and comprehensible wording, however, some technical corrections are necessary.

Specific comments:

1. Page 4, section 2.2: Please comment your choice. Why did you use those three

reanalyses?

2. Page 5, line 24-27: Why did you define blocked days only for selected regions? Please add a comment (pointing to Sect. 4.2).

3. Page 6, line 30: Do you have a possible explanation why the differences are larger near the maximum in the Euro-Atlantic blocking region?

4. Page 7, line 1-7: Please make clearer that the underestimation of the RO data (mentioned in the previous paragraph) is also visible in the time series of the SH blocking frequency.

5. Page 7, line 8-14: There were blocking events which lasted more than 30 days. One important example is the blocking event in summer 2010 which caused a mega heat wave in Russia and floods in Pakistan (e.g. Trenberth and Fasullo, 2012; Barriopedro et al., 2011; Hong et al., 2011). Since blockings show fluctuations in intensity during their life-cycle, you do not always get a continuous blocking signal with the common blocking indices. Thus, you are right, since you talk about "continuous" blocked days. However, you should try to make this point clearer. Some readers could wonder why such high impact blockings like the Russia heat wave block do not appear in your ranking.

6. Page 7, line 10-12: Are these blockings related to high impact weather events?

7. Page 10, line 5: This is of high relevance and points to the limit of the used data set/method. Emphasize this more strongly and point to possible consequences. This could be added somewhere in the next paragraph (line 8-12). Once again, the Euro-Russian summer block (2010) could be mentioned here (see comment 5.).

8. Page 13, line 35: Could you explain why?

9. Page 16, line 10: What about the signals at 850 hPa?

10. Page 16, line 29: Do you have an explanation why the largest anomalies are found at lower levels in the North Pacific region?

[Figure]

11. Page 18, section 5: Please compare your results with existing literature. It is not clear to the reader if your findings about the vertical structure of temperature and moisture anomalies are completely new. Does your findings (dis)agree with results from other studies?

Technical corrections:

12. Page 1, line 1: "high pressure" instead of "high-pressure".

13. Page 1, line 11: "equatorward" instead of "equator-ward".

14. Page 1, line 12: "anticyclonic" instead of "anti-cyclonic".

15. Page 2, line 1: Comma after "(NH)".

16. Page 2, line 7: Comma after "(SH)".

17. Page 2, line 7: Delete "about".

18. Page 2, line 9: Replace "Also" with (e.g.) "Furthermore".

19. Page 2, line 14: Please rephrase the sentence. Maybe replace one of the "and"s with "as well as".

20. Page 2, line 17: Comma after "Hence".

21. Page 2, line 24: Replace "Also" with "In addition".

22. Page 2, line 32: Comma after "Sect. 2".

23. Page 2, line 33: "Sect. 3" instead of "Section 3".

24. Page 5, line 9: Comma after "summer".

25. Page 5, line 9: "poleward" instead of "pole-ward".

26. Page 5, line 10: "equatorward" instead of "equator-ward".

27. Page 5, line 14: Replace "Simply speaking,..." with (e.g.) "To put it simple,...".
28. Page 5, line 21: Replace "too small" with "smaller".

29. Page 5, line 21: Comma after "step".

30. Page 6, line 22: Comma after "In the Euro-Atlantic region".

31. Page 6, line 27: Comma after "In the North Pacific region".

32. Page 7, line 4: Comma after "Pacific region".

33. Page 7, line 6: Comma after "150°W".

34. Page 7, line 12: Comma after "SH".

35. Page 7, line 13: Comma after "There".

36. Page 7, line 18: Reference after "ERA-Interim" to Figure 1.

37. Page 10, line 1: Comma after "analysis".

38. Page 10, line 6: Comma after "winter".

39. Page 10, line 17: Comma after "SH".

40. Page 10, line 21: "anticyclone" instead of "anti-cyclones".

41. Page 10, line 23: Comma after "SH".

42. Page 13, line 6: Add "(Fig. 5, left)" after "300 K".

43. Page 13, line 6: Delete "about" before "300 K".

44. Page 13, line 7: Delete "about" before "500 K".

45. Page 13, line 7: Replace "At lower pressures..." with "At upper levels,...".

46. Page 13, line 8: Delete "about" before "300 K" and add a comma.

47. Page 13, line 9: Delete "Higher up" and add comma after "stratosphere".

48. Page 13, line 10: Replace "Further up,..." with (e.g.) "At higher altitudes,...".

49. Page 13, line 14: Comma after "In the troposphere".

50. Page 13, line: 15: Add "left" after "Fig. 6".

51. Page 13, line 16: "anticyclonic" instead of "anti-cyclonic".

52. Page 13, line 16: "high pressure" instead of "high-pressure".

53. Page 13, line 26: Delete commas around "generally" and add one after "stratosphere".

54. Page 13, line 27: Replace "signal" with (e.g.) "influence".

55. Page 13, line 30: Add "(Fig. 6, left) after "summer".

56. Page 13, line 33: Add "(Fig. 6, right)" after "troposphere".

57. Page 13, line 35: Reduce "they appear higher up at about 200 hPa in summer" to "they appear at 200 hPa in summer".

58. Page 16, line 1: Add "left" after "Fig. 7".

59. Page 16, line 5: Comma after "stratosphere".

60. Page 16, line 7: Delete "about" before "500 hPa".

61. Page 16, line 8: Add "(Fig. 7, right)" after "lower pressures".

62. Page 16, line 8: Comma after "At 270 hPa and 200 hPa".

63. Page 16, line 9: Comma after "Above 200 hPa".

64. Page 16, line 11: Delete comma after "both".

65. Page 16, line 13: Add "(Fig. 8, left)" after "part of the troposphere".

66. Page 16, line 14: Comma after "tropopause".

67. Page 16, line 15: Delete "about" before "200 hPa".

68. Page 16, line 18: Add "regions" before "downstream".

69. Page 16, line 20: Add "(Fig. 8, right)" after "NH".

70. Page 16, line 20: Comma after "troposphere".

71. Page 16, line 22: Comma after "stratosphere".

72. Page 16, line 27: Delete "about" before "200 hPa".

73. Page 16, line 28: Delete "higher up".

74. Page 18, line 6: Comma after "(SH)".

75. Page 18, line 9: Comma after "During extended winter".

76. Page 18, line 10: Replace "Above about 500 hPa..." with "Above 500 hPa,...".

77. Page 18, line 11: Delete "about" before "270 hPa".

78. Page 18, line 12: Delete "about" before "200 hPa".

79. Page 18, line 12: Replace "Higher up..." with "Above 200 hPa,...".

80. Page 18, line 13: Comma after "troposphere".

81. Page 18, line 14: "anticyclonic" instead of "anti-cyclonic".

82. Page 18, line 14: Comma after "stratosphere".

83. Page 18, line 15: "equatorward" instead of "equator-ward".

84. Page 18, line 19: "equatorward" instead of "equator-ward".

85. Page 18, line 21: Delete "about" before "200 hPa".

---

## Referee Comment (RC2) · Anonymous Referee #2 · 26 Oct 2017

**General comments**

The paper "A global perspective on atmospheric blocking using GPS radio occultation – one decade of observations" by Brunner and Steiner analyses 10 years of blocking events as detected in GPS Radio Occultation (RO) data. Climatologies of blocking events as detected in RO data and in reanalysis data are studied: frequency as function of longitude and hemisphere, time evolution of blocking, and seasonal characteristics. The impacts of blocking on vertically resolved atmospheric temperature and specific humidity are investigated.

[Figure]

This paper presents an interesting application that utilizes the high vertical resolution of RO measurements - which is a unique feature of RO amongst satellite observation techniques. Global statistical studies of blocking have up to now been using models or reanalyses. Here, it is quite convicingly shown by Brunner and Steiner that RO data can be used as a fully observation-based alternative to the models.

The study seems to be an extension of a previous study by Brunner et al. [Atmos. Chem. Phys, 2016] in which the fundamental blocking detection technique is developed. The new study provides a 10-year climatological view, based on the previously developed techniques.

It is a well-written, clear, and concise report of the work undertaken. It is an interesting example of RO data contributing within a rather mature field of meteorology and atmospheric sciences. And, as also pointed out in the paper, it is a field that recently gained renewed attention due to its coupling to extreme weather, heat waves, etc.

It also seems that the potential shortcomings of the RO method for this particular application (mostly due to a low horizontal resolution due to under-sampling) are pointed out and at least partly explained.

The paper is well suited for publication in AMT.

**Specific comments - only minor**
Abstract, line 11: "equator-ward" should be "equatorward". Search for these, there are several of them in the text. Also "pole-ward" which should be "poleward".

Abstract, line 12: "anti-cyclonic" should be "anticyclonic". Search for this throughout the text.

Section 2.1: Which RO missions are included in the analysis? I don't find that information in this section. Perhaps it is found in one of the references.

Section 4.1: A specific day and grid cell is defined as either blocked (if certain conditions described in Secion 3.1 are met) or not blocked. If I understand it right, Figure 1 shows the overall frequency of blocking (i.e, Nblocked/Ntotal) for the whole 10-year period. What does "annual" blocking frequency mean in this context? I assume it means that data from all seasons are included, but to me "annual" indicates that data are separated into years.

Section 4.1, line 29: what does "one-dimensional" blocking frequency mean?

References, line 19, Brunner and Steiner: "amtospheric" should be "atmospheric"

---

## Author Comment (AC1) · 16 Nov 2017

**We thank the reviewer for the constructive and detailed review of our manuscript and for providing many helpful comments and technical corrections. Please find our detailed response to the comments in the following. The responses are highlighted in bold below each comment.**

General comments: The paper is well-structured, making it easy for the reader to follow. Title and abstract are excellent as they reflect the content of the paper quite well. Figures have a high quality. The results are represented well, however, a detailed comparison to results of previous studies is missing, especially in the summary and conclusion part. In addition, there are some statements that should be clarified. The paper is written in a clear and comprehensible wording, however, some technical corrections are necessary.

Specific comments:

1. Page 4, section 2.2: Please comment your choice. Why did you use those three reanalyses?

**We used three of the more recent reanalyses (in particular MERRA-2 only recently became available) which compare well against each other in terms of temperature and zonal winds (e.g., Long et al. 2017) to investigate the representation of blocking in reanalyses and compare it to RO.**

**We included this information and rephrased the respective sentence in Sect. 2.2, paragraph 1:**

**"Here, we selected three of the more recent reanalyses (Table 1), which compare well against each other, e.g., in terms of temperature and zonal winds (e.g., Long et al. 2017), to investigate their representation of blocking in comparison to RO: the European Centre for Medium-Range Weather Forecasts (ECMWF) Reanalysis Interim (ERA-Interim), the Japanese 55-year Reanalysis (JRA-55) by the Japan Meteorological Agency (JMA), and the recently published second Modern-Era Retrospective analysis for Research and Applications (MERRA-2) by the National Aeronautics and Space Administration (NASA)."**

2. Page 5, line 24-27: Why did you define blocked days only for selected regions? Please add a comment (pointing to Sect. 4.2).

**We added an explanation to make our approach clearer. The respective paragraph at the end of Sect. 3.1 now reads:**

**"To investigate the effects of blocking on temperature and humidity we further define blocked days with respect to three selected regions. A blocked day is found if at least one grid point is blocked in such a region. The regions are chosen to cover the areas of blocking maxima in both hemispheres. These main blocking regions are, in the following, referred to as the North Atlantic region (30°W to 10°E and 30°N to 72.5°N), the North Pacific region (160°E to 160°W and 30°N to 72.5°N), and the East Pacific region (150°W to 90°W and 72.5°S to 30°S). The coincidence of temperature and humidity anomalies during blocked days is tested statistically (see Sect. 3.2) in order to investigate the effects of blocking on the atmospheric temperature and humidity structure (c.f. Sect. 4.2)."**

3. Page 6, line 30: Do you have a possible explanation why the differences are larger near the maximum in the Euro-Atlantic blocking region?

**As Fig. 1 (left) shows, in the Euro-Atlantic blocking region, the blocking maximum in RO is shifted slightly to the east with respect to the maximum in reanalyses, which results in a larger difference in the area of the reanalyses maxima but a smaller difference east of it. However, from our study we are currently not able to determine the reason for that.**

4. Page 7, line 1-7: Please make clearer that the underestimation of the RO data (mentioned in the

previous paragraph) is also visible in the time series of the SH blocking frequency.

**We added a comment stating the underestimation of RO in the SH at the beginning of paragraph 2 in Sect. 4.1.It now reads:**
**"The right panel of Fig. 1 shows the SH blocking distribution. Again, all data sets agree on the main blocking region in the South Pacific, with RO again showing a slight underestimation of about 0.5%."**

5. Page 7, line 8-14: There were blocking events which lasted more than 30 days. One important example is the blocking event in summer 2010 which caused a mega heat wave in Russia and floods in Pakistan (e.g. Trenberth and Fasullo, 2012; Barriopedro et al., 2011; Hong et al., 2011). Since blockings show fluctuations in intensity during their life-cycle, you do not always get a continuous blocking signal with the common blocking indices. Thus, you are right, since you talk about "continuous" blocked days. However, you should try to make this point clearer. Some readers could wonder why such high impact blockings like the Russia heat wave block do not appear in your ranking.

**We thank the reviewer for this important suggestion. We added a sentence to make this clear. The respective paragraph in Sect. 4.1 now reads:**
**"The most persistent and continuous blocking cases in the NH occurred in March/April 2007 (27 days), in December/January 2009/2010 (28 days), and in February 2015 (23 days). All three cases were connected to unusual temperature anomalies, as e.g., discussed by Cattiaux et al. (2010) for winter 2009/2010 with severe cold spells hitting Europe.**
**Note that blocking can show considerable fluctuations in intensity during its evolution so that blocking cases may be interrupted by a few unblocked days and are not regarded as continuous signal. An example is the sequence of blockings in summer 2010 (see e.g., Brunner et al., 2016, Fig. 3) leading to a severe heatwave in Russia (e.g., Barriopedro et al., 2011)."**

6. Page 7, line 10-12: Are these blockings related to high impact weather events?

**The March/April 2007 blocking lead to unusually high temperatures for the season in the UK (e.g., https://www.metoffice.gov.uk/climate/uk/summaries/2007/april; accessed 02/11.2017), while the February 2015 blocking was connected to cold temperature particularly during night (e.g., https://www.metoffice.gov.uk/climate/uk/summaries/2015/february; accessed 02/11/2017). Blocking occurrence in winter 2009/2010 and its impacts are discussed, e.g., by Cattiaux et al. (2010).**
**We added this information in the paragraph in question, it now reads:**
**"All three cases were connected to unusual temperature anomalies, as e.g., discussed by Cattiaux et al. (2010) for winter 2009/2010 with severe cold spells hitting Europe."**

7. Page 10, line 5: This is of high relevance and points to the limit of the used data set/method. Emphasize this more strongly and point to possible consequences. This could be added somewhere in the next paragraph (line 8-12). Once again, the Euro-Russian summer block (2010) could be mentioned here (see comment 5.).

**The reviewer is right that the underestimation of the blocking maximum over northern Russia in summer is a shortcoming of the current RO data. We explicitly discuss the reasons for the underestimation of blocking with the current RO record in the following paragraphs. In the conclusions section (Sect. 5, last but one paragraph), we furthermore discuss upcoming RO missions and near-future constellations and the increase in the number of RO measurements (see e.g., Yue et al. 2014), potentially solving the undersampling of blocking in future.**

8. Page 13, line 35: Could you explain why?

**The reviewer raises an interesting point. In fact the anomalies near 200 hPa are quite similar for both seasons, while the anomalies at lower altitudes are weaker in summer. A possible explanation is that we compute the anomalies relative to the seasonal climatology and that, in**

general, tropospheric humidity is higher in summer than in winter. Anomalies in summer might thus be less pronounced than in winter.
**We rephrased the respective paragraph, it now reads:**
**"Specific humidity anomalies during summer blocking in this region are not statistically significant in most of the troposphere. Stronger anomalies are only visible near the tropopause and above, between 270 hPa to 200 hPa."**

9. Page 16, line 10: What about the signals at 850 hPa?
**We added a comment mentioning also 850 hPa, the sentence now reads:**
**"Specific humidity anomalies are strongest in the lower and middle troposphere between 850 hPa to 500 hPa and about 10% lower than in the North Atlantic region between 270 hPa and 200 hPa."**

10. Page 16, line 29: Do you have an explanation why the largest anomalies are found at lower levels in the North Pacific region?
**Specific humidity anomalies in the troposphere correlate strongly with the temperature anomaly. In the North Pacific region temperature anomalies are stronger at lower levels, compared to other regions. This might lead to the high specific humidity anomalies in this region.**

11. Page 18, section 5: Please compare your results with existing literature. It is not clear to the reader if your findings about the vertical structure of temperature and moisture anomalies are completely new. Does your findings (dis)agree with results from other studies?
**A focus of our study is the investigation of the atmospheric structure during blocking events in the RO data set. To our knowledge the RO data set has so far not been used for blocking diagnostics except in a first demonstration study by Brunner et al. (2016). However, the atmospheric response in general (e.g., positive temperature anomalies located in the region of the block and negative temperature anomalies on the eastern and southern flanks in the troposphere) is well known and is, e.g., described by Bieli et al. 2015.**
**We included additional references to recent studies at two places in Sect. 5:**
**"In the troposphere cold anomalies surround the central warm anomaly, indicating the effect of advection of cold air from the polar region by the anticyclonic motion around blocking highs which is in agreement with findings by Bieli et al. (2015)."**
**"However, the anomalies do not change sign at the tropopause, leading to inverse patterns of temperature and specific humidity in the lower stratosphere. This behavior of temperature and specific humidity anomalies at the tropopause level has recently also been noted by Sitnov et al. (2017)."**

Technical corrections:

12. Page 1, line 1: "high pressure" instead of "high-pressure".
**Done.**

13. Page 1, line 11: "equatorward" instead of "equator-ward".
**We replaced "equator-ward" by "equatorward" throughout the manuscript.**

14. Page 1, line 12: "anticyclonic" instead of "anti-cyclonic".
**We replaced "anti-cyclonic" by "anticyclonic" throughout the manuscript.**

15. Page 2, line 1: Comma after "(NH)".
**Done.**

16. Page 2, line 7: Comma after "(SH)".
**Done.**

17. Page 2, line 7: Delete "about".
**Done.**

18. Page 2, line 9: Replace "Also" with (e.g.) "Furthermore".
**Done.**

19. Page 2, line 14: Please rephrase the sentence. Maybe replace one of the "and"s with "as well as".
**We think the reviewer refers to page 2, line 16, where we replaced one "and". The sentence now reads:**
**"The systematic and global detection and analysis of atmospheric blocking as well as its impacts set demanding requirements to the data sets in use."**

20. Page 2, line 17: Comma after "Hence".
**Done.**

21. Page 2, line 24: Replace "Also" with "In addition".
**Done.**

22. Page 2, line 32: Comma after "Sect. 2".
**Done.**

23. Page 2, line 33: "Sect. 3" instead of "Section 3".
**We follow the AMT Manuscript preparation guidelines in our use of Section/Sect. and therefore prefer to leave this as it is. https://www.atmospheric-measurement-techniques.net/for_authors/manuscript_preparation.html: "The abbreviation 'Sect.' should be used when it appears in running text and should be followed by a number unless it comes at the beginning of a sentence."**

24. Page 5, line 9: Comma after "summer".
**Done.**

25. Page 5, line 9: "poleward" instead of "pole-ward".
**Done.**

26. Page 5, line 10: "equatorward" instead of "equator-ward".
**Done.**

27. Page 5, line 14: Replace "Simply speaking,..." with (e.g.) "To put it simple,...".
**Done.**

28. Page 5, line 21: Replace "too small" with "smaller".
**Done.**

29. Page 5, line 21: Comma after "step".
**Done.**

30. Page 6, line 22: Comma after "In the Euro-Atlantic region".

**Done.**

31. Page 6, line 27: Comma after "In the North Pacific region".
**Done.**

32. Page 7, line 4: Comma after "Pacific region".
**Done.**

33. Page 7, line 6: Comma after "150W".
**Done.**

34. Page 7, line 12: Comma after "SH".
**Done.**

35. Page 7, line 13: Comma after "There".
**Done.**

36. Page 7, line 18: Reference after "ERA-Interim" to Figure 1.
**We are not referring to Fig. 1 in this context but to the differences in two-dimensional blocking frequencies (not shown in the manuscript).**

37. Page 10, line 1: Comma after "analysis".
**Done.**

38. Page 10, line 6: Comma after "winter".
**Done.**

39. Page 10, line 17: Comma after "SH".
**Done.**

40. Page 10, line 21: "anticyclone" instead of "anti-cyclones".
**Done.**

41. Page 10, line 23: Comma after "SH".
**Done.**

42. Page 13, line 6: Add "(Fig. 5, left)" after "300 K".
**Done.**

43. Page 13, line 6: Delete "about" before "300 K".
**We prefer to keep "about" here.**

44. Page 13, line 7: Delete "about" before "500 K".
**We prefer to keep "about" here.**

45. Page 13, line 7: Replace "At lower pressures..." with "At upper levels,...".
**Done.**

46. Page 13, line 8: Delete "about" before "300 K" and add a comma.
**We write now: "Beginning near 300 hPa, "**

47. Page 13, line 9: Delete "Higher up" and add comma after "stratosphere".

**Done.**

48. Page 13, line 10: Replace "Further up,..." with (e.g.) "At higher altitudes,...".
**Done.**

**Additional references:**
**Barriopedro, D., Fischer, E. M., Luterbacher, J., Trigo, R. M., and García-Herrera, R.: The Hot Summer of 2010: Redrawing the Temperature Record Map of Europe, Science, 332, 220–224, doi:10.1126/science.1201224, 2011.**

**Cattiaux, J., Vautard, R., Cassou, C., Yiou, P., Masson-Delmotte, V., and Codron, F.: Winter 2010 in Europe: A cold extreme in a warming climate, Geophys. Res. Lett., 37, L20704, doi:10.1029/2010GL044613, 2010.**

**Long, C. S., Fujiwara, M., Davis, S., Mitchell, D. M., and Wright, C. J.: Climatology and interannual variability of dynamic variables in multiple reanalyses evaluated by the SPARC Reanalysis Intercomparison Project (S-RIP), Atmos. Chem. Phys. Discuss., pp. 1–43, doi:10.5194/acp-2017-289, 2017.**

**Sitnov, S., Mokhov, I., and Lupo, A.: Ozone, water vapor, and temperature anomalies associated with atmospheric blocking events over Eastern Europe in spring - summer 2010, Atmos. Environ., 164, 180–194, doi:10.1016/j.atmosenv.2017.06.004, 2017.**

---

## Author Comment (AC2) · 16 Nov 2017

**We thank the reviewer for the positive assessment and helpful comments. Please find our responses to the comments highlighted in bold below.**

General comments

The paper "A global perspective on atmospheric blocking using GPS radio occultation – one decade of observations" by Brunner and Steiner analyses 10 years of blocking events as detected in GPS Radio Occultation (RO) data. Climatologies of blocking events as detected in RO data and in reanalysis data are studied: frequency as function of longitude and hemisphere, time evolution of blocking, and seasonal characteristics. The impacts of blocking on vertically resolved atmospheric temperature and specific humidity are investigated.

This paper presents an interesting application that utilizes the high vertical resolution of RO measurements - which is a unique feature of RO amongst satellite observation techniques. Global statistical studies of blocking have up to now been using models or reanalyses. Here, it is quite convicingly shown by Brunner and Steiner that RO data can be used as a fully observation-based alternative to the models.

The study seems to be an extension of a previous study by Brunner et al. [Atmos. Chem. Phys, 2016] in which the fundamental blocking detection technique is developed. The new study provides a 10-year climatological view, based on the previously developed techniques.

It is a well-written, clear, and concise report of the work undertaken. It is an interesting example of RO data contributing within a rather mature field of meteorology and atmospheric sciences. And, as also pointed out in the paper, it is a field that recently gained renewed attention due to its coupling to extreme weather, heat waves, etc. It also seems that the potential shortcomings of the RO method for this particular application (mostly due to a low horizontal resolution due to under-sampling) are pointed out and at least partly explained. The paper is well suited for publication in AMT.

Specific comments - only minor

Abstract, line 11: "equator-ward" should be "equatorward". Search for these, there are several of them in the text. Also "pole-ward" which should be "poleward".
**We replaced "equator-ward" by "equatorward" and "pole-ward" by "poleward" throughout the manuscript.**

Abstract, line 12: "anti-cyclonic" should be "anticyclonic". Search for this throughout the text.
**We replaced "anti-cyclonic" by "anticyclonic" throughout the manuscript.**

Section 2.1: Which RO missions are included in the analysis? I don't find that information in this section. Perhaps it is found in one of the references.
**Information on which missions are specifically included, is found in the references cited in Table 1 for each reanalysis, respectively. For ERA-Interim, data from CHAMP, FORMOSAT-3/COSMIC, GRACE, MetOp, and TerraSAR-X are included (Poli et al. 2010, Table 1; Dee, 2016). For JRA-55, data from CHAMP, SAC-C, FORMOSAT-3/COSMIC, GRACE, MetOp, TerraSAR-X, and C/NOFS are included (Kobayashi et al. 2015, Appendix A). For MERRA-2, data from CHAMP, FORMOSAT-3/COSMIC, MetOp, GRACE, SAC-C, and TerraSAR-X are**

**included (see McCarty et al. 2016, Section 2.3.2).**
**We added the following information at the end of Sect. 2.2:**
**"All three reanalyses assimilate RO data. ERA-Interim includes measurements from CHAMP, COSMIC, GRACE, MetOp, and TerraSAR-X (Poli et al., 2010; Dee, 2016), MERRA-2 additionally includes SAC-C (McCarty et al., 2016), and JRA-55 all the former plus C/NOFS (Kobayashi et al., 2015)."**

Section 4.1: A specific day and grid cell is defined as either blocked (if certain conditions described in Secion 3.1 are met) or not blocked. If I understand it right, Figure 1 shows the overall frequency of blocking (i.e, Nblocked/Ntotal) for the whole 10-year period. What does "annual" blocking frequency mean in this context? I assume it means that data from all seasons are included, but to me "annual" indicates that data are separated into years.
**As the reviewer correctly assumes "annual" indicates that all seasons are included. This is consistent with the wording of the IPCC 2013 report (AR5; Box.14.2, Figure 1) where the phrase "Annual mean blocking frequency" is used to indicate that all seasons are included.**
**To make this clear, we now refer to "annual mean blocking frequency" in the caption of Fig.1 and in the related manuscript text in Sect. 4.1.**

Section 4.1, line 29: what does "one-dimensional" blocking frequency mean?
**We thank the reviewer for raising this question. The definition of one-dimensional blocking frequencies was indeed not included in the methods section.**
**In the revised version we state the definition of "one-dimensional" blocking in Sect. 3.1, last but one paragraph:**
**"Reducing the longitude-latitude view, one-dimensional blocking frequencies consider a given longitude in the northern or southern hemisphere as blocked if at least one latitude is blocked."**

References, line 19, Brunner and Steiner: "amtospheric" should be "atmospheric"
**We corrected it.**

**Additional references:**
**Dee, D., Fasullo, J., Shea, D., Walsh, J., and National Center for Atmospheric Research Staff: The Climate Data Guide: Atmospheric Reanalysis: Overview & Comparison Tables, https://climatedataguide.ucar.edu/climate-data/atmospheric-reanalysis-overview-comparison-tables, accessed November 15th, 2017, 2016.**